# Molecular Dissection of Phagocytosis by Proteomic Analysis in *Entamoeba histolytica*

**DOI:** 10.3390/genes14020379

**Published:** 2023-01-31

**Authors:** Natsuki Watanabe, Kumiko Nakada-Tsukui, Tomoyoshi Nozaki

**Affiliations:** 1Department of Biomedical Chemistry, Graduate School of Medicine, The University of Tokyo, Tokyo 113-0033, Japan; 2Department of Parasitology, National Institute of Infectious Diseases, Tokyo 113-0033, Japan

**Keywords:** phagocytosis, *E. histolytica*, proteomics

## Abstract

*Entamoeba histolytica* is the enteric protozoan parasite responsible for amebiasis. Trophozoites of *E. histolytica* ingest human cells in the intestine and other organs, which is the hallmark of its pathogenesis. Phagocytosis and trogocytosis are pivotal biological functions for its virulence and also contribute to the proliferation of nutrient uptake from the environment. We previously elucidated the role of a variety of proteins associated with phagocytosis and trogocytosis, including Rab small GTPases, Rab effectors, including retromer, phosphoinositide-binding proteins, lysosomal hydrolase receptors, protein kinases, and cytoskeletal proteins. However, a number of proteins involved in phagocytosis and trogocytosis remain to be identified, and mechanistic details of their involvement must be elucidated at the molecular level. To date, a number of studies in which a repertoire of proteins associated with phagosomes and potentially involved in phagocytosis have been conducted. In this review, we revisited all phagosome proteome studies we previously conducted in order to reiterate information on the proteome of phagosomes. We demonstrated the core set of constitutive phagosomal proteins and also the set of phagosomal proteins recruited only transiently or in condition-dependent fashions. The catalogs of phagosome proteomes resulting from such analyses can be a useful source of information for future mechanistic studies as well as for confirming or excluding a possibility of whether a protein of interest in various investigations is likely or is potentially involved in phagocytosis and phagosome biogenesis.

## 1. Introduction

*Entamoeba histolytica* is the enteric protozoan parasite that causes amoebiasis in humans. Phagocytosis and trogocytosis (simply expressed as phagocytosis, indicating both processes hereinafter) play a crucial role in the biology and pathogenesis of *E. histolytica*. *E. histolytica* trophozoites ingest dead cells to remove apoptotic and necrotic corpses by phagocytosis and invade into host tissue [1]. Trogocytosis is needed for killing host cells and cross-dressing (decorating its own surface with incorporated human cell membrane proteins) to protect itself from lysis by human serum [2]. A number of proteins involved in the process of phagocytosis have been identified and characterized toward the elucidation of a cascade of events during phagocytosis. However, since proteins and lipids involved in phagocytosis are regulated (i.e., recruited to and dissociated from the phagocytic site and the phagosome) in a spatiotemporal manner, it is difficult to have a global view of an entire panel of phagosomal proteins. In order to elucidate the mechanisms of phagocytosis, it is essential to carefully examine a panel of proteins identified by different biochemical and cell biological approaches. In this short review, we revisited our previous proteomic studies conducted in the last fifteen years and took advantage of the diversity of the methods and conditions used in these studies. These proteome data of phagosomes, which were isolated and purified using different protocols, were independently analyzed and never comprehensively discussed in the past. In this review, we made a list of proteins that were identified by all five independent studies of phagosome proteomics using latex beads as bait and buoyancy floating for phagosome isolation and also from magnetic isolation, using paramagnetic serum-coated beads as bait. These two methods allow us to differentiate the proteins strongly associated with phagosomes and those only transiently or weakly associated. Note that in the protocol using paramagnetic serum-coated beads, cells that internalized beads were treated by a cross-linker prior to pulling down, and thus, transiently or weakly associated proteins were also detected. The comprehensive analysis summarized in this review allowed us to identify common denominators, including Rab small GTPases, particularly the ones conserved among eukaryotes, but not those specific to Entamoeba, hydrolytic degrading enzymes, such as cysteine proteases, from all five sets of phagosome proteomes. Furthermore, Rho, filopodin, Hgl of Gal/GalNAc lectin, V-ATPase, and SyntaxinB were also reproducibly detected from phagosomes, irrespective of isolation protocols. In contrast, the proteins that are likely associated with phagosomes only transiently, including cytoskeletal proteins (actin-related protein 2/3 complex subunit, myosin heavy chain, F-actin-capping protein subunit α) and membrane traffic-related proteins [Vacuolar protein sorting (Vps) 29, Vps26, Vps35, adaptor protein family proteins] were identified only from the phagosomes with chemical crosslinking. Furthermore, some proteins, including α-amylase, cysteine protease 2, and myosine IB, were identified more abundantly by specific repression of phagocytosis-associated proteins, such as Atg8 or cysteine protease binding proteins (CPBF6), as well as under the condition of phosphatidylinositol 3-phosphate (PI3P) being depleted. These lists should be used as references for various cell biology studies, such as organelle and exo- (and ect-) some isolation, as well as protein–protein interaction studies. Altogether, our list of phagosome proteomes should not only serve as the baseline information to elucidate molecular mechanisms of phagocytosis in this parasite but be a useful reference to phagosome proteomics of other phagocytic pathogens as well as professional phagocytes of multicellular organisms.

## 2. Two General Methods for Phagosome Isolation from *E. histolytica* Trophozoites

To purify the phagosomes from *E. histolytica* trophozoites, two methods have been used: a density-gradient method using carboxylated latex beads [1,2] and magnetic isolation of phagosomes using tosyl-activated paramagnetic beads coated with the human serum [3,4,5,6]. In our laboratory, phagosome proteomic analysis has been repeatedly performed several times by using both methods (Table 1). In general, a lower number of proteins were detected from phagosomes purified by the density-gradient floating centrifugation method compared to those identified by using the magnetic isolation method (with the criteria for selection as phagosome proteins being the quantitative value (QV) over 0.7). In the previous review [1], we offered a summary based on one set of phagosome proteome data using phagosomes purified by the density-gradient method. This method can collect only the phagosomal proteins which are strongly and stably associated with phagosomes. On the other hand, by the magnetic-bead method, we could collect the proteins that were weakly or transiently associated with phagosomes. Table 1 and Table 2 summarize the Proteome 1–3 [1,2,6] or Proteome 4 and 5 of the control reference strain, but not overexpressed, or gene silenced strains of any particular genes. These data suggest two possibilities: the density-gradient centrifugation method yields isolated phagosomes with higher purity, or the method may fail to identify phagosome-associated proteins that either transiently or loosely bind to phagosomes. This is most likely due to chemical cross-linking by dithiobis (DSP) used in the paramagnetic-bead protocol, which allows the copurification of weakly associated proteins together with phagosomes. Furthermore, paramagnetic beads were also coated with human serum, which may induce the recruitment of serum component-dependent phagosomal proteins. Needless to say, the proteins which were not detected in the control (mock control of each experiment) were removed. Forty-one proteins were commonly identified in six independent phagosome proteomic studies and categorized into several functional groups.

## 3. Phagosomal Proteins Detected in All Proteomic Studies Represent a Core Set of Constitutive Proteins Necessary for Phagosome Biogenesis

Forty-one proteins were commonly detected in all phagosome proteome surveys irrespective of the methods of isolation of phagosomes and the presence or absence of the crosslinking reagent and serum coating of the beads. They are classified into eight categories (Appendix A). Among them, small G-proteins and lysosomal degrading enzymes represent more than a quarter (eleven) of 41 proteins. Small G-proteins are a family of GTP-binding proteins of 20–30 kDa and are conserved throughout eukaryotes. They are present in two forms, GTP-bound active and GDP-bound inactive forms, and play a pivotal role in the signaling of many cellular functions. There are five subfamilies, Ras, Rho, Rab, Afr, and Ran. In the list of the 41 proteins, three kinds of small G-proteins, eight Rab, two Rho, and one Ras, were found. Small G-proteins that belong to the Rho family regulate cell motility via cytoskeleton reorganization. Rab family G-proteins are involved in membrane trafficking between cellular compartments, such as the trans-Gogi network (TGN)–endosome, endosome–plasma membrane (PM), PM–endosome, and endosome–lysosome. Lysosomal hydrolytic degrading enzymes are known to be central to the pathogenesis and pathophysiology of amebiasis and also essential for digesting internalized prey, such as intestinal bacteria. Therefore, it is plausible that the small G-proteins, including Rho, Rab, and degrading enzymes, were commonly detected by all proteomic studies and play fundamental roles in phagocytosis and phagosome biogenesis.

Small G-proteins detected in all five studies are listed in Table 2. Several Rab proteins, which have been previously demonstrated to be involved in endocytosis, phagocytosis, and CP secretion, and also implicated in the destruction of mammalian cells, are included (Rab7A, Rab7D, and Rab11B) [8,9,10]. As stated above, Rab proteins are essential for vesicular transport (i.e., membrane traffic). Homo sapiens has around 60 Rab genes, while *E. histolytica* has more than 100 Rab genes [9]. The list contains eight Rabs including Rab7A, 7B, 7D, 11B, 11C, 1A, C1, and C3. Rab7A is known as the regulator of lysosome functions, particularly CP activity in the *E. histolytica* [8]. Thus, Rab7A is involved in phagocytosis via the regulation of phagosome maturation and lysosome fusion to phagosomes. Rab7A was also suggested to be involved in the degradation of transferrin in the lysosomes [11]. Rab7B is involved in the late-phase phagosome maturation in a coordinated fashion with Rab7A [12]. Rab5 and Rab7A are localized on pre-phagosomal vacuole (PPV), which is the preparatory compartment that emerges upon host cell attachment and demonstrated only in *E. histolytica* until today, and controls phagosome acidification by fusion to phagosomes in the later stage of phagosome biogenesis [13]. It was previously shown that Rab5 is associated with the PPV and replaced with Rab7A soon after PPV maturation prior to fusion with the phagosome [13]; thus, it is reasonable that Rab5 was not detected in Proteomes 1-3. Rab7D is also localized on lysosomes or PPV and inhibits phagocytosis when Rab7D is overexpressed [9].

Rab11B is partially associated with non-acidified vesicles, such as recycling endosomes. Intracellular and secreted cysteine protease activity is increased by Rab11B overexpression. Overexpression of Rab11B also enhanced exocytosis of the fluid-phase marker [10]. Rab11C expression increases throughout encystation, suggesting Rab11C is involved in the cyst formation [14]. However, the role of Rab11C in phagosome function in trophozoites and during encystation remains elusive. RabC1 and RabC3 were detected from all phagosome proteomes. It may be worth noting that *E. invadens* has two isotypes of RabC3, while *E. histolytica* has a single isotype [15]. However, no experimental data is available for the RabC family so far. Rab8A is known to be involved in the transport of a surface receptor or receptors from the ER to the PM [16] via binding to Cdc50 and a presumable lipid flipase P4 ATPase in *E. histolytica* [17]. If Rab8A is mainly engaged with biosynthetic and secretory pathways, it is reasonable that Rab8 was not detected from Proteomes 1-3.

Lysosomal digestive enzymes (listed in Table 2) are essential for the decomposition of internalized particles and substances by phagocytosis or micropinocytosis and are implicated in the pathogenesis of *E. histolytica* [18,19,20]. It has been established that expression levels of cysteine proteases correlate well with the apparent virulence [21]. CP-A1, CP-A2, and CP-A5 are known as the major CPs in *E. histolytica* and are important for tissue destruction and invasion and induction of macrophage proinflammatory response [22]. Importantly, CP-A1 and CP-A5 are absent from *E. dispar,* which is a non-virulent sibling species and cannot invade tissues and manifest disease symptoms [21]. EhCP2 is involved in the chemotaxis of leucocytes, i.e., modulation of leucocyte migration, by chemokine cleavage [15]. It has also been demonstrated that the expression level of CP-A4 in *E. histolytica* during amoebic liver abscess (ALA) formation was increased, suggesting that CP-A4 is associated with pathogenesis in the liver [23]. CP-A1, CP-A4, and CP-A5 were detected from all phagosome proteomes in *E. histolytica*. While the causal connection between the absence of CP-A1 and CP-A5 and the loss of apparent virulence in *E. dispar* is still elusive, the phagosome maturation (e.g., acidification) and degradation are significantly different between two sibling species as experimentally demonstrated [24,25]. The precise role of individual CPs in phagosome biogenesis remains unknown.

Lysozymes, α-amylases, and β-hexosaminidases (Table 2) were shown to be involved in the degradation of ingested bacteria in phagosomes of professional phagocytes from mammals [26,27,28]. In *E. histolytica,* lysozyme I, II, α-amylase, and β-hexosaminidase have been demonstrated to bind to the cysteine protease binding protein family (CPBF8) in *E. histolytica*. CPBF proteins are a family of eleven transporting receptors of lysosomal hydrolytic enzymes, including cysteine proteases, amylases, hexosaminidases, and lysozymes [29]. CPBF6 and CPBF8 are the most highly expressed CPBFs in *E. histolytica* trophozoites. Both CPBF6 and 8 are localized on phagosomes and lysosomes. CPBF8 gene silencing abolished transport of a panel of lysosomal enzymes to phagosomes reducing the destruction of CHO cells and ingestion of bacteria [30], suggesting that those transporters may also mediate signaling from phagosomes to the cell surface where adherence occurs prior to ingestion.

Glycoprotein (Gp) 63 is known as a surface protein from *Leishmania major* and, as integrated into exosomes, is involved in the evasion from host defense; thus, the crucial virulence factor [31,32,33] (Table 2). Gp63 suppresses the expression of the receptors of natural killer (NK) cells which assist in the immune response towards T helper type 1 in coordination with interferon-γ [34]. Gp63 is also known to affect the host immune systems via a selection of cargoes of exosomes. It was shown that inflammation and immune modulation were not controlled when Gp63 knockout *L. donovani* strain infected mice. Gp63 from *L. donovani* changes the expression level of surface proteins in NK cell [35]. In *E. histolytica*, two proteins are encoded by Gp63 genes (Gp63-1, EHI_200230; Gp63-2, EHI_042870). The expression level of Gp63-1 is approximately 300 times higher than Gp63-2. Gp63-1 was detected from all phagosome proteomes, while Gp63-2 was detected from only one proteome with a low quantitative value (proteome No. 4, under GFP-FYVE expressing condition). Gp63-1 was also detected from exosomes from *E. histolytica* [36] (Santos *et al*., unpublished data). Identification of Gp63-1 in the core phagosome proteome indicates two possibilities: (1) Gp63-1 is involved in phagosome biogenesis; (2) the phagosome may have cross-talk with the late endosomes and the multivesicular bodies in which the exosomes are likely formed.

## 4. Phagosome Proteins Detected Exclusively by the Magnetic Bead Isolation Method, Combined with Bead Serum Coating and Chemical Cross-Linking, Represent Those Involved in Serum-Dependent Phagocytosis or Transiently Associated with Phagosomes

A panel of phagosome proteins was exclusively identified by the magnetic bead isolation method, combined with serum coating of beads and chemical cross-linking (Proteome 4 and 5). We believe that the two lists of phagosomal proteins created based on two protocols show functional and temporal (i.e., strength and timing of the interaction with phagosomes) differences in their association with phagosomes. As stated above, a few key differences between the two methods are: (1) paramagnetic beads were coated by a human serum to mimic target human cells; (2) beads were collected by either magnetic separation using paramagnetic beads or by density gradient centrifugation; (3) chemical cross-linking was applied to the magnetic isolation method. Proteomes 4 and 5 (using the paramagnetic beads isolation method) detected twice to thrice as many proteins as Proteomes 1–3, which were gained by the density-gradient centrifugation protocol. The average number of proteins of Proteomes 1–3 was 227, while that of Proteomes 4 and 5 was 811 (Table 1). Two hundred thirty-one phagosome proteins were differentially detected in Proteomes 4 and 5 but not in Proteomes 1, 2, and 3 (Appendix A), and were categorized by PANTHER (http://pantherdb.org/about.jsp, accessed on 21 December 2022) (Figure 1). One hundred and fifty-eight out of 231 proteins were classified into thirteen groups. The largest group, “metabolite interconversion enzyme”, included two long-chain-fatty-acid-CoA ligases, phosphatidylserine synthase, phosphoglycerate kinase, and diacylglycerol O-acyltransferase. Six out of thirty proteins are lipid metabolism-related enzymes (Appendix A). The second largest groups are the “membrane traffic protein” and “translational protein” categories. The third to fifth largest groups are the “protein-binding activity modular”, “protein modifying enzyme”, and “cytoskeletal protein” categories, respectively.

The largest group of “metabolite interconversion enzymes” includes the enzymes involved in phospholipid metabolism (EHI_009800: phosphatidylserine synthase; EHI_068320: glycerophosphoryl diester phosphodiesterase, EHI_070720: inositol-3-phosphate synthase), a glycolytic enzyme (EHI_188180: phosphoglycerate kinase), and general lipid metabolism-related enzymes (EHI_113590: diacylglycerol O-acyltransferase; EHI_079300 and EHI_188590: long-chain-fatty-acid-CoA ligase) were detected. These lipid metabolism- and (phospho) lipid metabolism-related enzymes identified in Proteomes 4 and 5 suggest that these metabolic pathways are transiently or weakly associated with phagosomes during phagosome biogenesis, as previously suggested [37]. Lipid metabolic enzymes may be needed for the elongation of the phagosomal membrane.

In the “membrane traffic protein” category, vacuolar protein sorting (Vps) 26-1 (EHI_062490, Watanebe et al., unpublished), 29 (EHI_025270), 35-1, and 35-2 (EHI_002990, EHI_041950) were detected from Proteomes 4 and 5. These proteins make a retromer complex, which is known to be important for the retrograde transport of hydrolase receptors from lysosomes (endosomes) to the TGN and the surface transmembrane receptors from endosomes to the PM [38]. Vps26 localization on the phagosome in *E. histolytica* was confirmed by immunofluorescence assay [4,8]. The finding that the retromer components were exclusively identified from phagosomes after beads serum coating and cross-linking is consistent with the notion that the interaction of the retromer with phagosomes is transient and time-dependent. Vps26 was characterized as a Rab7A effector protein and co-localized with Rab7A [8]. Rab7A is localized on the phagosome stably; however, Vps26 localized with it transiently. This fact may be caused by the alteration of the GTP and GDP of Rab7A. Vps26 can bind to GTP type Rab7A, which suggests that Rab7A transiently changes to GTP Rab7A and Vps26 binds to it at that time.

In addition to six *E. histolytica* Rabs that have orthologs in other species and are broadly conserved in eukaryotes (Rab1A, 7A, 7B, 7D, 11B, and 11C) and two Entamoeba-specific Rabs (RabC1 and C3), six additional Entamoeba-specific Rabs (RabA, RabC5, RabC7, RabK1, RabI2, and RabP2) (note that Entamoeba-specific Rabs are named primarily in alphabetic order, followed by a numerical order for isotypes), were detected only from Proteomes 4 and 5. RabC7 and RabP2 were also detected in an independent proteome study [3] using serum-coated magnetic beads. It suggests that these two Rabs may be recruited only when the bait is coated with human serum. RabA has been shown to be involved in motility, polarization [15,39], and transport of the Gal/GalNAc lectin [15,40]. According to the phylogeny reconstruction of Entamoeba Rabs, after *E. histolytica*-specific Rabs (RabX), including RabX16, 17, and 32, branched off the main tree, the RabC group was formed. RabC group was then divided into two subgroups, RabC1/C3 and RabC5/C7 [15]. It may be worth rephrasing that RabC1 and RabC3 are constitutively (or tightly) associated with phagosomes, whereas similar isotypes RabC5 and RabC7 are weakly or transiently associated. RabI is evolutionarily relatively close to the Rab7 subfamily, indicating that RabI may also play an important role in phagosome biogenesis similar to Rab7 isotypes. Note that RabP2 is present in *E. dispar* but absent in *E. invadens* [15]. The role and localization of RabK1 and RabI2 remain uncharacterized. Rab21 is known to be involved in endocytosis and phagocytosis in general [15,41]. Rab21 is found in Proteomes 4 and 5 (Appendix A); however, this is a canonical type of Rab that is conserved in eukaryotes.

Twenty-two proteins were categorized into “translational proteins”. These proteins include ribosomal proteins, translation initiation factors, and tRNA synthetases, which are localized in ribosomes, and potentially associated with the ER. These ER-localized proteins were also detected in phagosomes from mammalian cells. The phagosome and the ER form a membrane contact site (MCS), which is needed to transport the ER-resident proteins to the phagosomes [42]. Therefore, the detection of these proteins from phagosomes is reasonable.

“Protein modifying enzyme” includes seventeen proteins. Five out of seventeen are ubiquitin-proteasome-related proteins. The ubiquitin-proteasome system is known as a selective proteolytic pathway and is different from autophagy, which is a non-selective protein degradative pathway. However, it has been reported that the ubiquitinated proteins are also delivered via vesicles to the late endosome and are degraded by the lysosome [43]. It is not clear whether this ubiquitinated lysosome degradation pathway is conserved in *E. histolytica*. The discovery of the ubiquitin–proteasome system in Proteomes 4 and 5 suggests that the ubiquitin–proteasome system interacts with phagosomes or, alternatively, the system is entirely engulfed by autophagosomes in *E. histolytica*. The interaction between ubiquitin–proteasome system and phagosomes has not been explored in this organism.

Hemolysin, which is categorized as a “transmembrane signal receptor”, is usually secreted from bacteria, and has been shown to degrade red blood cells and induce apoptosis of immune cells. The role of hemolysin in *E. histolytica* remains controversial, while the gene is used as the target for diagnostic PCR [44].

PANTHER analysis indicates that sixteen proteins are categorized into “cytoskeletal proteins”. These proteins include actin-related protein, actin-binding protein, actin, and F-actin-capping protein. Actin is known to be involved in many essential cellular processes of *E. histolytica*, such as motility, pseudopod formation, adherence, endo- and exocytosis, and phagocytosis [45,46]. In *E. histolytica*, 390 actin-binding proteins (ABPs) were identified by in silico analysis based on their domain arrangements [45]. In humans, only 162 ABPs are encoded, suggesting that *E. histolytica* ABPs are more diversified within the species [47]. ABPs are known to diversify the function of actin by modifying actin functions, as actin is extremely conserved throughout eukaryotes’ function. Therefore, six ABPs found in Proteomes 4 and 5 may modulate actin participation during phagosome biogenesis in a lineage-specific fashion. Actin and phagosome-associated ABPs may be involved in pathogenesis via phagocytosis-related functions. Rho1 and Rho7 were detected from all phagosome proteomes (Table 1). This result supports the importance of Rho-regulated actin rearrangement during phagocytosis and phagosome biogenesis. Furthermore, filopodin, one of the actin-related proteins, was detected in all phagosome proteomes.

“RNA metabolism proteins” and “chromatin binding proteins“ are involved in translation and gene expression. These proteins were unexpectedly detected from phagosome proteomes. The proteins detected from phagosomes and isolated by the magnetic bead method were categorized by PANTHER (Appendix A) and subsequently, analyzed by STRING (https://string-db.org) (Appendix A). STRING analysis indicates that the proteins that belong to “RNA metabolism proteins” and “chromatin-binding proteins” have relationships with ribosomal proteins or chaperons, which are localized in the ER. It is well known, as stated above, that phagosomes and the ER make a physical interaction via MCS [42].

Furthermore, proteins categorized into “metabolite interconversion enzyme” were unexpectedly detected from phagosome proteomes 4 and 5. STRING analysis also indicates that a good number of proteins that are categorized into “metabolite interconversion enzymes” were also linked to ER localizing proteins. For example, EHI_188180 (phosphoglycerate kinase), a glycolytic enzyme, is linked to a transcriptional protein (EHI_125650) and ER ATPase (cell division cycle 48: EHI_045120). KEGG analysis (https://www.genome.jp/kegg/pathway.html, accessed on 21 December 2022) of the proteins of “metabolite interconversion enzyme” indicates predominant roles of phagosome-associated metabolism in “Carbohydrate metabolism” (12 proteins) and “Lipid metabolism” (8 proteins) (Appendix A). These observations are consistent with the role of phagosomes as a nutrient provider.

## 5. Conclusions

In this review, we summarized all the phagosome proteome data that had been created using two different phagosome isolation methods in our own laboratory and categorized them into two major categories: the core constitutive proteome and the proteome of transiently or weakly associated constituents (Appendix A). The catalog of the phagosome proteomes should be a useful reference to easily validate whether proteins of interest are involved in phagosome biogenesis directly or indirectly. As far as we are concerned, this is the first review to systematically compare phagosome proteomes obtained by different methods in a single laboratory. By comparing data obtained by two methods, we gained a new overall picture of phagosome proteomes. For example, broadly conserved canonical Rabs were constitutively detected from all phagosome proteomes, while Entamoeba-specific Rabs six were detected only by magnetic isolation method combined with cross-linking, suggesting that Entamoeba-specific Rabs may access to the phagosomes only transiently and weakly, and presumably control the multiple fate of phagosomes, which well illustrates diversified vesicular trafficking mechanisms in *E. histolytica*. Other than expected players, unexpected proteins, including Gp63 and lipid metabolism enzymes, were detected from phagosomes. Furthermore, a panel of cysteine proteases and hydrolases reinforces their central role in phagosome biogenesis. “Metabolite interconversion enzyme” is one of the largest categories. This may suggest that many unknown metabolic enzymes/pathways interact with phagosomes during phagosome biogenesis.

## Figures and Tables

**Figure 1 genes-14-00379-f001:**
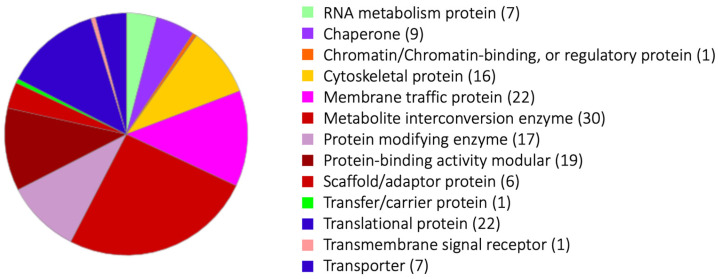
Gene ontology analysis of phagosome proteins isolated by the magnetic bead isolation method. The pie chart of thirteen “protein classes” of the proteins detected from the phagosomes collected by the magnetic bead isolation method. The list of proteins is in Appendix A. The numbers after the categories in Figure 1 are detected protein numbers.

**Table 1 genes-14-00379-t001:** The list of phagosome proteome performed in our laboratory. Phagosomes were purified by two different methods: the density-gradient and magnetic-bead methods. In a magnetic-bead method, beads were coated with human serum, and the samples were treated with DSP for cross-linking.

Number	Journal	Year	Method	Detected Proteins	Reference
1	*Mol. Biochem. Parasitol.*	2006	Density-gradient	155	[2]
2	*Arch. Med. Res.*	2006	Density-gradient	151	[1]
3	*Infect. Immun.*	2013	Density-gradient	375	[6]
4	*Cell. Microbiol.*	2019	Magnetic beads (serum coated; cross linking)	884	[4]
5	*Front. Cell. Infect. Microbiol.*	2022	Magnetic beads (serum coated; cross-linking)	738	[5]

One phagosome proteomic study from our laboratory [7] was not included in this review because the proteome was largely identical to what was described in [2].

**Table 2 genes-14-00379-t002:** Small G-proteins and degrading enzymes detected in all proteomes, and Small G-proteins detected in Proteomes 4 and 5 and categorized into “protein-binding activity modulators”.

AmoebaDB Annotation	EHI Number	Common Name	Phagosome Isolation Method	Proteome Number	General Function
Rab family GTPase	EHI_108610	Rab1A	Density-gradient/ Magnetic beads	1,2,3,4,5	Membrane traffic
Small GTPase Rab7A	EHI_192810	Rab7A	Density-gradient/ Magnetic beads	1,2,3,4,5	Early phagosome maturation
EhRab7B protein	EHI_081330	Rab7B	Density-gradient/ Magnetic beads	1,2,3,4,5	Membrane traffic
EhRab7D protein	EHI_082070	Rab7D	Density-gradient/ Magnetic beads	1,2,3,4,5	Phagosome maturation
Small GTPase Rab11B	EHI_107250	Rab11B	Density-gradient/ Magnetic beads	1,2,3,4,5	Phagosome maturation (in Eh)/CP secretion (in Eh)/endosome recycling
Rab family GTPase	EHI_161030	Rab11C	Density-gradient/ Magnetic beads	1,2,3,4,5	Endosome recycling
Rab family GTPase	EHI_153690	RabC1	Density-gradient/ Magnetic beads	1,2,3,4,5	Membrane traffic
Rab family GTPase	EHI_143650	RabC3	Density-gradient/ Magnetic beads	1,2,3,4,5	Membrane traffic
Rho family GTPase	EHI_070730	Rho1	Density-gradient/ Magnetic beads	1,2,3,4,5	Cell motility
Rho family GTPase	EHI_129750	Rho7	Density-gradient/ Magnetic beads	1,2,3,4,5	Cell motility
Ras family GTPase	EHI_058090		Density-gradient/ Magnetic beads	1,2,3,4,5	Signal transduction
Cell surface protease gp63, putative	EHI_200230	Gp63	Density-gradient/ Magnetic beads	1,2,3,4,5	Degrading enzyme
Lysozyme, putative	EHI_199110	Lysozyme I	Density-gradient/ Magnetic beads	1,2,3,4,5	Degrading enzyme
Lysozyme, putative	EHI_096570	Lysozyme II	Density-gradient/ Magnetic beads	1,2,3,4,5	Degrading enzyme
Cysteine proteinase 1	EHI_074180	CP-A1	Density-gradient/ Magnetic beads	1,2,3,4,5	Degrading enzyme
Histolysain	EHI_033710	CP-A2	Density-gradient/ Magnetic beads	1,2,3,4,5	Degrading enzyme
Cysteine proteinase, putative	EHI_050570	CP-A4	Density-gradient/ Magnetic beads	1,2,3,4,5	Degrading enzyme
Cysteine proteinase	EHI_168240	CP-A5	Density-gradient/ Magnetic beads	1,2,3,4,5	Degrading enzyme
Dipeptidyl-peptidase	EHI_136440		Density-gradient/ Magnetic beads	1,2,3,4,5	Degrading enzyme
α-amylase family protein	EHI_023360		Density-gradient/ Magnetic beads	1,2,3,4,5	Degrading enzyme
β-hexosaminidase	EHI_007330		Density-gradient/ Magnetic beads	1,2,3,4,5	Degrading enzyme
Serine carboxypeptidase (S28) family protein	EHI_054530		Density-gradient/ Magnetic beads	1,2,3,4,5	Degrading enzyme
Rab family GTPase	EHI_168600	RabA	Magnetic beads	4,5	Membrane traffic
Rab family GTPase	EHI_079890	RabC7	Magnetic beads	4,5	Membrane traffic
Rab family GTPase	EHI_122730	RabC5	Magnetic beads	4,5	Membrane traffic
Rab family GTPase	EHI_053420	RabI2	Magnetic beads	4,5	Membrane traffic
Rab family GTPase	EHI_024680	RabK1	Magnetic beads	4,5	Membrane traffic
Rab family GTPase	EHI_117890	RabP2	Magnetic beads	4,5	Membrane traffic
Ras-related protein	EHI_129330	Rab21	Magnetic beads	4,5	Membrane traffic
Rho family GTPase	EHI_190440	Rho10	Magnetic beads	4,5	Cell motility
Rho family GTPase	EHI_135450	Rho13	Magnetic beads	4,5	Cell motility
Ras family GTPase	EHI_198330	Not assigned	Magnetic beads	4,5	Signal transduction

## Data Availability

Not applicable.

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
