# Peer review of "Molecular Dissection of Phagocytosis by Proteomic Analysis in Entamoeba histolytica"

_genes, 2023, doi:10.3390/genes14020379_

Round 1

Reviewer 1 Report

This manuscript focuses on a review of the findings related to the proteins that have been identified by two different methodologies in the process of phagocytosis in the Entamoeba histolytica parasite. Previous research articles where proteins were identified are compared to determine their role in phagocytosis and/or phagosome biogenesis. Although it is an important review for the field, some concerns are enlisted as follows:

Mayor considerations:

1. As phagocytosis is an important virulence property that characterizes E. histolytica, authors should give a broader overview of the process and its biological importance in the parasite.

2. Tables 2, 3, and 4 presented in the review do not give enough information and are not homogenous. A suggestion could be to fusion in one table divided by functions and add two columns, one with the extraction method in which it was identified and the number of assays and the other with a brief description of the function in which the protein participates example. "immune evasion", "endosome recycling", "late phagosome maturation". The table could include the reference information.

3. Figure 1 title must be changed, the authors are showing a gene ontology analysis, and the number of proteins identified in each category could be shown. The authors could include the different levels of the GO analysis and statistical analysis.

4. Supplementary materials are of great relevance in this review however, they are not adequately described in the text or the file. Also, proteins in supplemental table 2 must be organized through functions, tables 1 and 3. Titles must be changed for a better description as well as inclusion throughout the review.

5. A STRING analysis related to the different categories obtained by PANTHER could indicate the relevance of the protein interactions and associate them with a particular function.

6. There is not enough discussion of the unexpected proteins found in the proteomic assays such as RNA metabolism proteins, chromatin binding proteins, and metabolite interconversion enzymes. This information is interesting and a KEGG analysis of the metabolite interconversion enzyme could help give an interpretation of these results and the importance of their space-temporal presence in the phagosome. Because the review intends to enlighten and molecularly dissect the process of phagocytosis instead of mostly describing already reported and common phagocytosis proteins.

7. Finally, the work mostly emphasizes a comparison of previous findings through verbal description, however, the inclusion of a scheme, figure, or model could give a better understanding of the process and enrich the review.

Minor consideration.

1. There are some misspelled words in the manuscript such as "phogosomes" in line 60 page 2 and "E. histoltyica" in line 137 page 4

2. Authors use "exclusively identified only" in line 170 page 5 authors must eliminate one of them.

3. Some italic words are missing like in "Homo sapiens" line 114 page 3. And in some references.

Author Response

  1. As phagocytosis is an important virulence property that characterizes  histolytica, authors should give a broader overview of the process and its biological importance in the parasite.

 >> Thank you for your advice. I added new sentences about phagocytosis and trogocytosis as follow:

E. histolytica trophozoites ingest dead cells to remove apoptotic and necrotic corpses by phagocytosis and invade into host tissue [1]. Trogocytosis is needed for killing host cells and cross dressing (decorating its own surface with incorporated human cell membrane proteins) to protect itself from lysis by human serum [2].” at line 33-36.

  1. Tables 2, 3, and 4 presented in the review do not give enough information and are not homogenous. A suggestion could be to fusion in one table divided by functions and add two columns, one with the extraction method in which it was identified and the number of assays and the other with a brief description of the function in which the protein participates example. "immune evasion", "endosome recycling", "late phagosome maturation". The table could include the reference information.

>> Thank you for the suggestion. We combined three tables to make one and added three columns to indicate “Phagosome isolation method”, “Proteome number”, and “General function”. We also modified the table number refered in the main text.

  1. Figure 1 title must be changed, the authors are showing a gene ontology analysis, and the number of proteins identified in each category could be shown. The authors could include the different levels of the GO analysis and statistical analysis.

>>Thank you for the suggestion. We changed the title of Figure 1 as follows:

“Gene ontology analysis of phagosome proteins isolated by the magnetic beads isolation method”. We also added the number of proteins of each category in parenthesis after the name of categories in Figure 1. Also, we added the p-value and the number of proteins in the category of PANTHER analysis for the statistical information in Supplemental table 3 (after the category name).

  1. Supplementary materials are of great relevance in this review however, they are not adequately described in the text or the file. Also, proteins in supplemental table 2 must be organized through functions, tables 1 and 3. Titles must be changed for a better description as well as inclusion throughout the review.

>> Thank you for the comment. We added the legends of all supplemental tables in the main text (line 362-372). We also changed the title of each table in the excel file. Supplemental table 3, which we also provided in the initial submission, was the reanalyzed PANTHER-categorized phagosomal proteins shown in Supplemental Table 2.

  1. A STRING analysis related to the different categories obtained by PANTHER could indicate the relevance of the protein interactions and associate them with a particular function.

>>Thank you for the advice.  We have done STRING analysis (https://string-db.org) and a result of the analysis was attached as Supplemental figure 1 in revision. The legend of Supplemental figure 1 was also added (line 373-376). Also, discussion of this analysis was added at line 319-337 (also see our response to comment #6)

  1. There is not enough discussion of the unexpected proteins found in the proteomic assays such as RNA metabolism proteins, chromatin binding proteins, and metabolite interconversion enzymes. This information is interesting and a KEGG analysis of the metabolite interconversion enzyme could help give an interpretation of these results and the importance of their space-temporal presence in the phagosome. Because the review intends to enlighten and molecularly dissect the process of phagocytosis instead of mostly describing already reported and common phagocytosis proteins.

>>Thank you for your comments and suggestions. We conducted STRING and KEGG analyses as explained above. Such analyses also included phagosome associated proteins related to RNA metabolism proteins, chromatin binding proteins, and metabolite interconversion enzymes. We carried out KEGG analysis (https://www.genome.jp/kegg/pathway.html) of the proteins that belong to “metabolite interconversion enzyme” and the result was attached as Supplemental Table 4.

We also added the following discussions (lines 319-337):

“RNA metabolism proteins” and “chromatin binding proteins“ are involved in translation and gene expression. These proteins were unexpectedly detected from phagosome proteomes. The proteins detected from phagosomes isolated by the magnetic beads method were categorized by PANTHER (Supplemental Table 3) and subsequently analyzed by STRING (https://string-db.org) (Supplemental Fig 1). STRING analysis in-dicates that the proteins that belong to “RNA metabolism proteins” and “chromatin binding proteins” have relationships with ribosomal proteins or chaperons which are localized in the ER. It is well known, as stated above, that phagosomes and the ER make a physical interaction via MCS [43].

Also, proteins categorized into “metabolite interconversion enzyme” were unex-pectedly detected from phagosome proteomes 4 and 5. STRING analysis also indicates that a good number of proteins that are categorized into “metabolite inter conversion enzyme” were also linked to ER localizing proteins. For example, EHI_188180 (phosphoglycerate kinase), a glycolytic enzyme, is linked to a transcriptional protein (EHI_125650) and ER ATPase (cell division cycle 48: EHI_045120). KEGG analysis (https://www.genome.jp/kegg/pathway.html) of the proteins of “metabolite intercon-version enzyme” indicates predominant roles of phagosome-associated metabolism in “Carbohydrate metabolism” (12 proteins) and “Lipid metabolism” (8 pro-teins)(Supplemental Table 4). These observations are consistent with the role of phago-somes as a nutrient provider.”

  1. Finally, the work mostly emphasizes a comparison of previous findings through verbal description, however, the inclusion of a scheme, figure, or model could give a better understanding of the process and enrich the review.

>> Thank you for the advice. In order to give an overview or summary of this review, we made a new venn diagram which illustrates the major categories and representative proteins from phagosome proteomes, as Supplemental Figure 2 (legend line 377-379). We hope that this supplementary figure makes it easy to visualize two major categories: the core constitutive proteome and the proteome of transiently or weakly associated constituents, described in Conclusion (Section 8). (line 342).

Minor consideration.

  1. There are some misspelled words in the manuscript such as "phogosomes" in line 60 page 2 and "E. histoltyica" in line 137 page 4

 >>Thank you for that. We modified these two word correctly.

  1. Authors use "exclusivelyidentified only" in line 170 page 5 authors must eliminate one of them.

>> We removed “only” from the sentense.

  1. Some italic words are missing like in "Homo sapiens" line 114 page 3. And in some references.

>> We modified them to italic words.

Reviewer 2 Report

Comments: Molecular dissection of phagocytosis by proteomic analysis in 2 Entamoeba histolytica

This review provides an excellent overview of current knowledge of proteins involved in phagocytosis and trogocytosis in Entamoeba histolytica. The authors have accurately and concisely summarized the research that has been conducted to date, and have provided a catalog of proteomes associated with phagosomes. The authors also provide a useful source of information for future studies by highlighting the core set of constitutive and condition-dependent proteins associated with phagosomes. The language used is clear and concise, and the manuscript is well-structured. Overall, this is a well-written review that provides a comprehensive overview of current research in this field.

Major:

1. Please provide a brief description in the table of the modifications made to the method that resulted in the identification of more proteins than before.

2. I would greatly appreciate if you could include the statistical values, such as the p-value, from the Panther software analysis in the Excel sheet.

 Minor:

1. Please check line 18 in the abstract section and correct the spelling mistake - "To date" instead of "to data". Also, please correct other spelling mistakes throughout the manuscript.

Author Response

  1. Please provide a brief description in the table of the modifications made to the method that resulted in the identification of more proteins than before.

 >> We added a brief description of the methods in the legend of Table 1 (line 101-103).

“Phagosomes were purified by two different methods: the density-gradient and magnetic beads methods. In magnetic beads method, beads were coated with human serum and the samples were treated with DSP for cross linking.”

  1. I would greatly appreciate if you could include the statistical values, such as the p-value, from the Panther software analysis in the Excel sheet.

>> Thank you for the suggestion. We added the p-value in Supplemental Table 3.

 Minor:

  1. Please check line 18 in the abstract section and correct the spelling mistake - "To date" instead of "to data". Also, please correct other spelling mistakes throughout the manuscript.

>> Thank you for pointing out it. We modified it.